# Embolization of Ruptured Infratentorial Pial AVM in Pregnancy

**DOI:** 10.3390/life13040896

**Published:** 2023-03-28

**Authors:** Kamil Zeleňák, Dušan Šalát, Branislav Kolarovszki, Egon Kurča, Jana Zeleňáková, Naci Koçer

**Affiliations:** 1Clinic of Radiology, Jessenius Faculty of Medicine in Martin, Comenius University in Bratislava, Kollárova 2, 03659 Martin, Slovakia; 2Faculty of Health Sciences, University of Ss. Cyril and Methodius in Trnava, Námestie J. Herdu 577/2, 91700 Trnava, Slovakia; 3Clinic of Neurosurgery, Jessenius Faculty of Medicine in Martin, Comenius University in Bratislava, Kollárova 2, 03659 Martin, Slovakia; 4Clinic of Neurology, Jessenius Faculty of Medicine in Martin, Comenius University in Bratislava, Kollárova 2, 03659 Martin, Slovakia; 5Department of Neuroradiology, Cerrahpasa Medical Faculty, Istanbul University-Cerrahpasa, 34320 Istanbul, Turkey; 6Department of Neuroradiology, Acibadem University Hospital Group, 34303 Istanbul, Turkey

**Keywords:** arteriovenous malformation, rupture, embolization, pregnancy, radiation, dose, EVOH, PHIL, brain

## Abstract

A primigravida 22-year-old woman, at a gestation of 23 weeks, experienced bleeding from a pial arteriovenous malformation (AVM) located in the right cerebellum. After interdisciplinary consensus and with the informed consent of the patient and her family, AVM embolization was performed. Complete occlusion of the AVM was achieved by embolization with PHIL (precipitating hydrophobic injectable liquid). The calculated dose in the uterus was less than 1 µSv, which represents a negligible risk of harmful effects on the fetus. She delivered a baby at 37 weeks of gestation by cesarean section without complications. No congenital disorders were diagnosed by standard screening methods until the age of the newborn was two years. The angiography protocol must be optimized to minimize the radiation dose. Adequate shielding protection of the uterus is important. Premature termination of pregnancy is not necessary. Multidisciplinary care of neurologists, neurosurgeons, interventional radiologists, anesthesiologists, neonatologists, and obstetricians is necessary.

## 1. Introduction

Brain arteriovenous malformations (bAVMs) are uncommon vascular lesions (the incidence is approximately 1 per 100,000 per year in unselected populations) composed of feeding arteries (usually enlarged), a nidus (with shunt), and draining veins. bAVMs are typically present in young adults with spontaneous intracranial hemorrhage, seizures, or headaches [1,2,3]. Brain arteriovenous malformations have been commonly regarded as congenital, but this suspected origin has been challenged by published reports of de novo bAVMs [4].

The prevalence rate of bAVMs is approximately 0.01–0.5%, and they generally present symptoms between 20 and 40 years of age, most commonly at 30 years of age, and equally affect men and women [5]. The location ratio of bAVMs is 85% vs. 15% for supratentorial (two-thirds superficial and one-third deep) vs. infratentorial [3].

The annual risk of bleeding from unruptured bAVMs was estimated from retrospective studies at 4%, but it has recently been reported in a range of 1–2.2% [6,7]. It was ≈2% in the observation arm of the ARUBA study [6]. The lifetime risk of hemorrhage from non-ruptured bAVMs can be simply approximated with the following formula: lifetime risk (%) = 105—the patient’s age in years [8]. The bleeding originates from the brain arteriovenous malformation itself or from the venous side [6]. The rebleeding rate is higher, especially in the first year after rupture, ranging from 6% to 15.8% [7]. Statistically significant risk factors for hemorrhage are previous hemorrhage, deep bAVM location, exclusive deep venous drainage, and associated aneurysm [2].

ICARUSS investigators positively confirmed a drainage vein diameter as a specific risk factor for AVM rupture with a cutoff ≥ 5 mm [9].

Compared to a nonpregnant period in women with AVMs, the potential risk of intracranial hemorrhage increases 3.27 times during pregnancy and puerperium [10]. Because of the high rate of rebleeding, the alarming maternal mortality rate from rebleeding (up to 28%), and the 14% fetal death rate, the treatment of ruptured bAVMs should not be delayed and must be actively managed [5]. The intervention of ruptured bAVMs is recommended within 2 weeks after initial bleeding in patients of gestational age < 34 weeks, while termination of pregnancy as soon as possible followed by timely intervention of ruptured bAVMs is practicable in patients of gestational age ≥ 34 weeks [11].

The main characteristics of brain AVMs are listed in Table 1.

## 2. Management of Radiation Dose Reduction

At zero doses, there is zero risk of harm. In general, the ALARA principle (As Low As Reasonably Achievable) assumes that exposure to radiation poses a risk. This fundamental principle is even more crucial to adhere to during pregnancy.

### 2.1. Basic Principles and Radiation Limits

The civil population is constantly exposed to radiation from natural sources. Today, the medical use of X-rays represents the largest source of man-made radiation. Selected radiation doses are listed in Table 2 [12,13].

There is no definitive evidence to support the risk at all doses. This remains an active area of research. Depending on the study, approximately 100 mSv, 50 mSv, or 10 s mSv may be quoted as a level below which evidence is difficult to determine [14]. Better predictions of low dose/dose rate effects that are of great concern in the field of radiation protection can be reached by using the negative binomial linear-quadratic model variants [15]. 

The European Directive 2013/59/Euratom provides a relevant and beneficial practical impact on European radiology [16]. The dose limits for occupational exposure apply to the sum of annual occupational exposures of a worker from all authorized practices listed in Table 3 [17].

#### Radiation Limit and Fetal Doses

The International Commission on Radiological Protection (ICRP) recommends that the maximum dose of radiation to the uterus during pregnancy be less than 100 mGy to minimize the teratogenic risks to the fetus [18,19].

The approximate fetal doses from radiological examinations are listed in Table 4 [20].

### 2.2. Radiation Dose Reduction and Protocol Optimization

It is extremely important to constantly train medical professionals on how to adequately reduce radiation associated with medical imaging techniques [21]. Operators must be familiar with the principles of how to reduce and measure radiation dose in fluoroscopically guided interventions [22,23,24,25,26,27,28,29] and protect themselves [30,31], especially pregnant women who are interventional radiologists [32]. Essential principles are listed in Table 5.

An absorbed dose threshold to the patient’s brain of 0.5 Gy is exceeded quite often during cerebral embolization procedures, and optimizing the protocol is crucial. Brain doses are almost doubled when the field increases from 8 to 15 cm or more [33].

It is necessary to avoid or at least minimize abdominal fluoroscopy in pregnant patients. The fetal limit can be approached if the fluoroscopy time exceeds 7 min [19]. The shielding of the abdomen with a lead skirt with a lead equivalent of 0.5 mm must be used in the pregnant patient during neurointerventional procedures.

#### New Technologies

Spot fluoroscopy represents an innovative approach to adequately reducing radiation dose in neurointerventional procedures that led to a reduction of 50% of the total fluoroscopic dose-area product [34].

A new generation of 76 µm pixel high-resolution high-definition (hi-def) zoom mode was developed for use in interventional procedures that require superior image quality over a small field of view [35].

## 3. Embolization Techniques

Endovascular treatment of brain arteriovenous malformations has evolved considerably over the last few decades. Adaptively, it represents an alternative therapy to and not only a useful adjuvant technique to conventional neurosurgery.

Enormous progress has been made in the development of liquid embolic materials and microcatheters. New polymers enable better penetration of an AVM nidus and a more excessive rate of complete AVM occlusion. Current commercially available liquid embolic materials are listed in Table 6.

The availability of detachable microcatheters optionally allows for long-time injection of embolic material. The distal tip (1.5 to 5 cm) of these microcatheters can be left naturally in the access vessel, and the proximal part of the catheter is carefully removed at the end of the treatment procedure. The unique combination of the new embolic material and the detachable microcatheter typically allows for the performing of control angiography during the complex procedure. The injection of embolic material can be interrupted and continued several times as needed.

New imaging techniques are highly needed for the selection of the optimal treatment strategy, a more precise visualization of the origin of the main draining vein of the bAVM, and the depiction of more accurate intranidal aneurysms [36,37].

### 3.1. Transarterial Embolization

Endovascular embolization by the transarterial approach is preferred in most bAVMs as a conventional technique. The purpose of this technique is to reduce intranidal pressure before occlusion of the drainage vein. Such embolization can be used as a single therapeutic procedure or as neoadjuvant therapy prior to microsurgery or stereotactic radiosurgery.

A possible combination with microsurgical resection can also be used for grade III–V bAVMs as well. A high rate of total obliteration was achieved: 96.8% (in 30/31) was achieved with relatively low rates of permanent morbidity and mortality [38].

Specific modifications of the transarterial technique traditionally represent the plug technique, the pressure cooker technique, the occlusion balloon technique, and multiple catheter techniques.

#### 3.1.1. Plug Technique and Prolonged Intranidal Injection

The plug mitigates the potential risk of backflow and catheter entrapment, thus the “reverse plug then push” technique, allowing the operator to inject higher volumes of embolic material at greater injection rates [39].

Detachable microcatheters allow for long-time injection. A higher rate (51%; 179/350) of AVM obliteration was achieved with prolonged intranidal injection compared to the technique previously used [40].

#### 3.1.2. Pressure Cooker Technique

The pressure cooker technique was designed to block the backflow of embolic material to the nontarget artery by quickly creating an antireflux plug by trapping the detachable part of the microcatheter with coils and glue [41].

The modified coil-protected transarterial n-butyl cyanoacrylate injection technique using a single microcatheter to typically prevent reflux can be efficiently preferred when rapid hemostasis and short procedure time are required in hemodynamically unstable conditions [42].

#### 3.1.3. Occlusion Balloon Technique

The dual-lumen balloon catheter technique was uniquely created to minimize the necessary procedure and fluoroscopy times to promptly form an adequate proximal plug that allows the forwarding of nidal penetration while preventing reflux and nontargeted embolization [43].

The present generation of modern devices (the Scepter Mini) can be additionally handled for superselective flow arrest and navigation support due to its ability to be correctly placed on small-diameter feeders [44].

Balloon catheters also play a substantial role in AVM embolization. Inflation of a compliant balloon in the parent artery immediately after the origin of the feeder is a simple but effective maneuver during microcatheter navigation [45].

#### 3.1.4. Multiple Catheter Technique

Occlusion of the draining veins before complete exclusion of the nidus typically carries a procedural risk of bleeding. Two or more arterial feeders can be catheterized and simultaneously embolized to achieve total embolization of the nidus before reaching the venous drainage [46,47].

### 3.2. Transvenous and Combined Techniques

Unlike the transarterial approach, curative embolization of AVMs can be achieved more easily with a transvenous approach. The proposed indications for this approach include a small (diameter < 3 cm) and compact AVM nidus, deep AVM location, hemorrhagic presentation, single draining vein, lack of an accessible arterial pedicle, exclusive arterial supply by perforators, and en passage feeding arteries [48].

Transvenous retrograde AVM nidus sclerotherapy under controlled hypotension (TRENSH) technique using temporary systemic hypotension with or without temporary occlusion of the main arterial feeders was first conceptualized by Massoud and Hademenos [49]. Extensive knowledge of the venous anatomy is crucial to prevent some of the potential complications of the procedure [50,51].

Hudak established a novel strategy of combined transarterial and transvenous embolization with a new liquid embolic agent based on pol-urethane. A combined transarterial and transvenous embolization technique was carefully applied for 46 high-grade bAVMs. Patients experienced subarachnoid or intracerebral hemorrhage (29/46), drug-resistant epilepsy, and/or severe focal deficit (17/46). The principal author precisely measured the local arterial and venous pressure of the nidus during the specific procedure. Transarterial embolization was initially used at the beginning of the complex procedure. After a sufficient reduction in the inflow to the nidus, temporary controlled systemic hypotension was typically induced. At the last moment, the new liquid embolic agent performed a proper transvenous penetration of the nidus. Despite the exclusive inclusion of high-grade bAVMs in the published study, the contributing author achieved complete occlusion of AVMs in 20 specific patients. In the remaining 26 cases, an occlusion rate of 80–90% was reasonably achieved [52]. The fundamental limitation of this study remains the fact that the embolic material used is commercially unavailable.

In most published data, the transvenous approach is typically used in extraordinary cases with the infeasibility of arterial embolization, including supply by tiny perforating arteries, feeders ‘in passage’, or no clear arterial pedicle, and/or in combination with other arterial approaches where the intention of the intervention was endovascular cure. There was an exceedingly higher rate of complete AVM obliteration (38/41; 92.6%) and excellent functional outcomes (0% procedure-related mortality and 2.5% overall mortality) in patients with ruptured and unruptured AVMs treated with transvenous embolization in another prospective study. The mean size of the AVM nidus was 2.8 ± 1.2 cm, and low Spetzler-Martin grade AVMs comprised 41.5% of the lesions. Most of the patients (23/41; 56%) were treated in one session. The new Dyna 4D technique was incorporated into the preprocedural analysis. The distinguished authors slowly injected Onyx and tolerated a maximum reflux of 3 cm. After completion of the operative procedure, the microcatheter was typically cut at the optimal level of the jugular sheath [53].

In the review of the literature of 13 studies (69 patients with 70 AVMs) on transvenous embolization, the complete AVM obliteration rate was reported in 93% of specific cases [48].

The transvenous retrograde pressure cooker technique using coils and n-butyl-2-cyanoacrylate as a venous plug can be curative if the main inclusion criteria (AVM < 3 cm and single main drain vein) are accepted without reserve [54].

Solely transvenous embolization can be used in selected small-sized ruptured bAVMs. Fusion of magnetic resonance imaging and 3D angiography is helpful in controlling and fully achieving bAVM penetration by the embolic agent [55].

Selective temporary flow arrest during transvenous endovascular embolization (TFATVE) represents a novel adaptation of the transvenous approach, which employs hyper-compliant balloons intraarterially for the careful blocking of arterial feeders at the time of injection of ethylene-vinyl copolymer to reduce intranidal pressure and increase nidus occlusion rates [56].

Modern dual microcatheterization techniques with arterial and/or venous access, graciously according to angioarchitecture, was curative for low-grade Spetzler-Martin (SM I vs. II was 22% vs. 78%) bAVMs in 95%. More than 60% of prospective patients had ruptured AVMs [57].

Combined embolization therapy appears to be safe and potentially curative for certain deep AVM, e.g., in the basal ganglia, insula, and thalamus. Such a specific location is extremely demanding for all treatment techniques, and microsurgery might not be indicated. Complication rates that may, fortunately, reach radiosurgery profiles were demonstrated in a cohort study of 22 patients. A single transarterial approach was used precisely in nine (41%) unique cases, double catheterization in four (18%) cases, and the transvenous approach was required in eight (36%) cases. The mean size of the AVM was 2.98 ± 1.28 cm, and the mean number of sessions was 2.1 per patient. Most of the patients had cerebral hemorrhage (18/21, 82%), and three (14%) patients had a deteriorated neurological state (modified Rankin Scale > 2) at presentation. Sixty-eight percent of ruptured AVMs had a size ≤ 3 cm [58].

The ongoing study of the transvenous approach for the treatment of cerebral arteriovenous malformations (TATAM) was designed to adequately test the hypothesis that transvenous embolization is superior to transarterial embolization for obliteration of arteriovenous malformations [59].

## 4. Case

A primigravida 22-year-old female with no medical history, at a gestation of 23 weeks, experienced bleeding from a pial arteriovenous malformation (AVM) located in the right cerebellum. She presented with acute-onset headache with accompanying meningeal and cerebellar syndrome, nausea, and vomiting. The patient underwent magnetic resonance imaging, which confirmed bleeding from the right cerebellar AVM.

Less than 2% of infants <24 weeks’ gestation and approximately half of infants from 24 to 27 weeks’ gestation survived to discharge home from the neonatal intensive care unit [60]. Compared to a nonpregnant period in women with AVMs, the potential risk of intracranial hemorrhage increases 3.27 times during pregnancy and puerperium [10]. Because of the high rate of rebleeding, the alarming maternal mortality rate from rebleeding (up to 28%), and the 14% fetal death rate, the treatment of ruptured bAVMs should not be delayed and must be actively managed [5]. Surgical treatment of posterior fossa AVMs is associated with a higher rate of complications. Most posterior fossa AVMs are associated with an increased hemorrhage rate. Thus, they are a predictor of a poor outcome and should be treated even if unruptured to maintain good neurological function [61]. The potential risk of maternal blood pressure instability during surgery may negatively affect the fetus. Cerebellar location and deep venous drainage are predictors of hemorrhage. Endovascular embolization is useful and safe for infratentorial AVMs [62].

All the mentioned arguments were discussed in detail during the interdisciplinary meeting. After interdisciplinary consensus and with the informed consent of the patient and her family, AVM embolization was performed. The procedure was performed under general anesthesia. The abdominal area of the patient was protected with the shielding of a lead skirt with a 0.5 mm lead equivalency.

The right common femoral artery was punctured under ultrasound guidance, and a 6F short sheath was placed by the Seldinger technique. The patient was not heparinized. Over the guidewire, the sheath was replaced by a 6F 80 cm long introducer, which was placed in the aortic arch without the use of fluoroscopy. Fluoroscopy was used only in the head and neck area of the patient. The procedure continued by positioning the introducer into the left subclavian artery. Then a 6F guiding catheter was advanced and positioned coaxially into the left vertebral artery. All catheters were flushed continuously with saline solution. The left vertebral artery angiogram confirmed the pial AVM located in the right cerebellum, which was supplied by three branches of the right superior cerebral artery and drain into the straight sinus (Figure 1). Apollo microcatheters (ev3 Neurovascular, Irvine, CA, USA), with a 3 cm long detachable tip length, were navigated into three supplying arteries by guidewires: Hybrid 008.J and Hybrid 007.J (Balt, Montmorency, France), Synchro-10/300 (Stryker Neurovascular, Fremont CA, USA), and Asahi 008/300 (Asahi Intecc, Aichi, Japan). Three feeding arteries were embolized one by one with 25% liquid embolic material PHIL (MicroVention, Tustin, CA, USA) in amounts of 1.55, 0.45, and 1.5 mL. The complete embolization of the AVM was achieved at the end of the procedure (Figure 1). The total procedure time was 1 h and 50 min. The total amount of Visipaque 320 iso-osmolar iodine contrast medium (GE Healthcare, Oslo, Norway) applied to the patient was 54.7 mL.

The Allura Clarity biplane interventional X-ray system (Philips, The Netherlands) was used. A low X-ray dose level mode was selected except for the time of critical embolization moments. A maximum of 12.5 frames per second fluoroscopy rate was used. The total fluoroscopy time was 33 min and 20 s. The dose in the uterus, calculated by the Monte Carlo program for calculating patient doses in medical X-ray examinations, PCXMC 2.0 software version, was less than 1 µSv. She delivered a baby at 37 weeks of gestation by cesarean section without complications. No congenital disorders were diagnosed by standard screening methods until the age of the newborn was two years. The patient recovered without a focal neurological deficit and returned to her normal life. The follow-up magnetic resonance confirmed total AVM occlusion.

## 5. Discussion

Ruptured pial AVM in pregnancy is a rare but extraordinarily complex situation, associated with high maternal and fetal mortality [5,11]. Most bAVMs occur in the supratentorial region [11], but, in our case, the AVM was located in the infratentorial region. This location of the AVM is usually more difficult to treat surgically. AVMs of the posterior fossa brain are twice as likely to cause hemorrhage compared to supratentorial AVMs. Cerebellar location and deep venous drainage are predictors of hemorrhage. Endovascular embolization is useful and safe for infratentorial AVMs [62]. Three feeding arteries were embolized one by one with liquid embolic material PHIL 25% in our case.

The International Commission on Radiological Protection (ICRP) recommends that the maximum dose of radiation to the uterus during pregnancy be less than 100 mGy to minimize the teratogenic risks to the fetus [18,19]. Extremely limited information on the dose of fetal radiation during embolization is available in the literature. Asano et al. reported an exposure level of the uterus of 0.14 mGy [63]. Although three feeding branches were embolized in our case in one session to achieve complete occlusion of the AVM, the dose in the uterus was less than 1 µSv. The fetal dose can be close to zero with proper shielding, and premature termination of the pregnancy is not necessary. The dose achieved in our case represents a negligible risk of harmful effects on the fetus.

Nagayama et al. reported an average exposure dose to the abdominal surface in the presence of lead sheets during stereotactic radiosurgery of 14.05 mGy, and the fetus was exposed to a radiation dose of 8.26 mGy [64]. Radiosurgery does not reduce the risk of rebleeding immediately after application. Radiosurgery has a limited indication for the treatment of ruptured AVMs in pregnancy. Yu reported that fetal radiation doses approximately corresponded to 0.01% of the maximum tumor dose in a pregnant patient who underwent stereotactic radiosurgery with a gamma knife [65].

Surgery must be indicated in active hydrocephalus requiring shunting, associated with signs of imminent herniation, progressive neurological deficit, or both [66].

Visipaque 320 (iodixanol) is not contraindicated in pregnancy [67]. Only 54.7 mL of iso-osmolar iodine contrast media and 3.5 mL of a new generation of liquid embolic material, precipitating the hydrophobic injectable liquid (PHIL) embolic agent, were applied. She delivered a baby at 37 weeks of gestation by cesarean section without complications. No congenital disorders were diagnosed by standard screening methods until the age of the newborn was two years. No teratogenic effects were observed. The patient recovered completely.

Different embolic materials were used for embolization during pregnancy, including NBCA, Glubran, and Onyx—Table 7. The iodine-based embolic material PHIL was selected for use in our case because of its unique ability to penetrate the AVM nidus and create fewer artifacts compared to tantalum. PHIL is believed to flow forward rather than like a column and with less lava-like behavior with precipitation from the outside to the inside [68].

There are no definitive guidelines on how to properly treat ruptured bAVMs during pregnancy. Only a few authors report the radiation dose to the fetus, which is a limit to creating a comparison group. Our center reports only one case. 

## 6. Conclusions

There are no definitive guidelines on how to properly treat ruptured bAVMs in pregnancy. The decision is challenging due to its rarity. Patients may benefit from early therapy due to a higher rate of excessive rebleeding during pregnancy. The purpose of treatment is to preserve the life of the mother as well as the viability of the fetus. Limited data on the dose of fetal radiation during embolization are published, but the available data are acceptable and smaller compared to radiosurgery. Embolization of ruptured AVMs during pregnancy can be used alone as a curative treatment or as an essential part of therapeutic management. Embolization of ruptured AVMs during pregnancy is a feasible treatment technique. The angiography protocol must be optimized to minimize the radiation dose. Adequate shielding protection of the uterus is important. In our case, the calculated dose in the uterus was less than 1 µSv, which represents a negligible risk of harmful effects on the fetus. Premature termination of pregnancy is not necessary. Multidisciplinary care involving neurologists, neurosurgeons, interventional radiologists, anesthesiologists, neonatologists, and obstetricians is necessary.

## Figures and Tables

**Figure 1 life-13-00896-f001:**
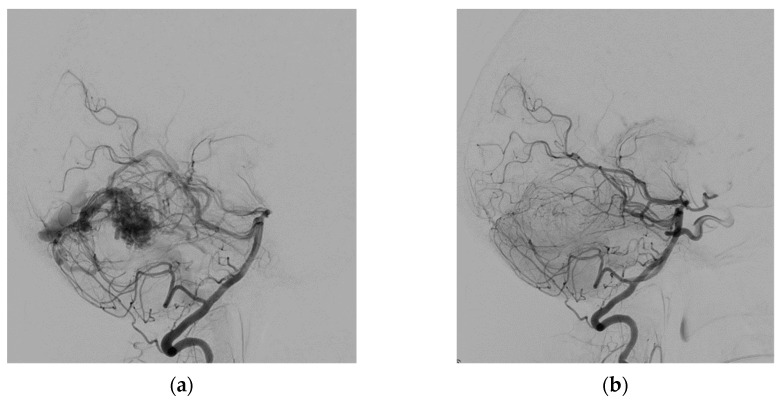
Angiogram of the left vertebral artery (lateral view) before (**a**) and after (**b**) AVM embolization.

**Table 1 life-13-00896-t001:** Characteristics of brain AVMs.

Brain AVM	Value
Incidence	Approximately 1 per 100,000 per year [1,2,3]
Prevalence Rate	Approximately 0.01–0.5% [5]
Location	85% vs. 15% for supratentorial (two-thirds superficial and one-third deep) vs. infratentorial [3]
The annual risk of bleeding from unruptured bAVM	Approximately 1–4% [6,7]
Rebleeding Rate	Is higher, especially in the first year after rupture, ranging from 6% to 15.8% [7]
The potential risk of intracranial hemorrhage compared to a nonpregnant period in women with AVM	Increases 3.27 times during pregnancy and puerperium [10]
Mortality Rate	Maternal mortality rate from rebleeding (up to 28%) and 14% fetal death rate [5]

**Table 2 life-13-00896-t002:** List of radiation dose examples.

Dose	Example
0.1 micro-Sievert (μSv)	The banana equivalent dose (BED)—is the dose one is exposed to by eating one average-sized banana because it contains radioactive isotopes of potassium ^1^
1 μSv	The average annual dose to a ‘heavy’ consumer of Irish Sea seafood ^1^
8 μSv	The dose received on a return flight from Dublin to London ^1^
20 μSv	Dose from a single chest X-ray ^1^ 1 in 1,000,000 lifetime risk of fatal cancer ^1^
0.4 mSv = 400 μSv	The worldwide average annual effective dose for the population from diagnostic medical X-ray examinations ^2^
2.4 mSv = 2400 μSv	The worldwide annual average effective dose from natural sources (6.57 μSv daily) ^2^

^1^ Health risk—Environmental Protection Agency [12], ^2^ UNSCEAR 2000 Report, Vol. 1 [13].

**Table 3 life-13-00896-t003:** List of dose limits for professional workers.

Dose	Example
1 milli-Sievert (mSv)	As soon as a pregnant worker informs the employer of the pregnancy, the employment conditions must ensure that the equivalent dose to the unborn child is as low as reasonably achievable and unlikely to exceed 1 mSv during at least the remainder of the pregnancy. ^1^
6 mSv	Apprentices and students (aged 16–18 years) have more restrictive dose limits: an effective dose of 6 mSv per year and equivalent doses of 15 mSv per year in the lens of the eye, as well as 150 mSv per year in the skin and extremities. ^1^
20 mSv	The limit on the effective dose shall be 20 mSv in any single year (in special circumstances 50 mSv if the average annual dose over any 5 consecutive years does not exceed 20 mSv). ^1^
20 mSv	The limit on the equivalent dose for the eye lens shall be 20 mSv in a single year or 100 mSv in any 5 consecutive years subject to a maximum dose of 50 mSv in a single year. ^1^
500 mSv	The limit on the equivalent dose for the skin and extremities shall be 500 mSv in a year. ^1^

^1^ European Society of Radiology [17].

**Table 4 life-13-00896-t004:** Approximate fetal doses from radiological examinations.

Fetal Dose	Examination
<0.01 mGy	Chest PA ^1^
0.1 mGy	CT chest ^1^
1 mGy	Abdomen or pelvis AP ^1^
1.5 mGy	Lumbar spine (AP and lateral) ^1^
25 mGy	CT pelvis ^1^
100 mGy	Maximum dose of radiation to the uterus during pregnancy recommended by ICRP ^2,3^

^1^ Table modified from [20], ^2,3^ [18,19].

**Table 5 life-13-00896-t005:** Protocol optimization to reduce the fetal dose.

Principles	Examples	Clinical Practice
Time	Limit fluoroscopy and exposure time	Select the most experienced operator
	Limit abdominal or pelvic fluoroscopy	The fetal limit can be approached if the fluoroscopy time exceeds 7 min
	Reduce the number of frames per second	Select the appropriate fluoroscopy mode
	Ultrasound use	Access artery puncture Guidewire navigation in the pelvic region
Distance	Keep the source-to-image distance (SID) low	Keep the detector as close to the patient as possible
	Distance between source and fetus	Limited application
	Important for operators mainly	Inverse square law
Shielding	Protect the abdominal area of the patient/fetus	Lead skirt shielding with a lead equivalent of 0.5 mm
	Collimate	Whenever possible
Other	Increase field of view (FOV) and collimate	Limit image magnification
	Prefer the lateral view	
	Minimize C-arm angulation	If you cannot reduce it, move it
	Experience	
New technologies	Spot fluoroscopy	50% reduction in the total fluoroscopic dose-area product
	High-definition detector	High-definition zoom mode

**Table 6 life-13-00896-t006:** List of liquid embolic material.

Material Characterization	Trademark	Company–Producer
N-Butyl-2-cyanoacrylate (n-BCA)	TRUFILL n-BCA liquid embolic system	Codman Neuro
Synthetic modified co-monomer cyanoacrylate glue	Glubran 2	GEM
n-hexyl-cyanoacrylate	Magic glue	Balt
Ethylene-vinyl alcohol (EVOH) copolymer	Onyx/Squid/Menox	Medtronic/Balt/Meril
Polylactide-coglycolide and polyhydroxyethylmethacrylate	PHIL (precipitating hydrophobic injectable liquid)	Microvention

**Table 7 life-13-00896-t007:** List of publications reporting embolization of bAVMs during pregnancy.

Author	Year	Number of Cases/Age of Pregnant Women (Year)	Gestational Age at the Time of AVM Rupture (Weeks)	Embolic Material/ Number of Sessions	AVM Size, cm/Grade/ Location	Radiation Dose to the Fetus	Baby Injury	Note
Trivedi [69]	2003	1/30	16	—	—/4/dominant frontal	—	No	Preoperative embolization
Granata [70]	2010	1/39	27	1 mL of Onyx 18/1	0.8/2/left parietal	—	No	Radiosurgery after pregnancy
Salvati [71]	2011	1/23	19	NBCA/1	—/—/right frontal	—	No	Complete AVM occlusion
Dashti [72]	2012	1/17	20	Onyx/2	5.5/IV/posterior fossa	1.9 × 10^−30^ mGy	No	The patient delivered healthy twin girls at 36 weeks of gestation by planned cesarean section
Jermakowicz [67]	2012	1/23	22	Onyx 18/1	—/—/Left P–O	—	No	50% embolization performed 3 weeks after bleeding, followed by complete resection 3 weeks later
Asano [63]	2016	1/34	25	50% NBCA/1	1/2/right occipital	0.14 mGy	No	Elective cesarean section at 38 weeks of gestation
Porras [73]	2017	4/25–41	19–38	—	2–7/3–4/R–P/O, L–P/O, R–F/P, R–O	—	No	In 3 of 4 embolizations combined with radiosurgery. C-section 29–40 weeks.
Sohail [74]	2019	1/20	15	20% Glubran/1	—/3/right hemisphere	—	—	Termination of pregnancy by hysterotomy
Teik [75]	2019	2/32–33	14–33	—/1–2	2–3.2/—/basal ganglia, left parasagital	—	No	A patient with basal ganglia AVM died with an unsuccessful pregnancy
Yan [76]	2021	3/16–26	16–23	—/1	1.1–7.6/—/Right T/O, left T, right cerebellar	—	No	Embolization and resection in all cases. One patient underwent two resections.

## Data Availability

Data are available from the authors upon reasonable request.

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
