# Peer review of "Embolization of Ruptured Infratentorial Pial AVM in Pregnancy"

_life, 2023, doi:10.3390/life13040896_

Round 1

Reviewer 1 Report

This case report describes the successful treatment of a pregnant patient with a ruptured infratentorial pial arteriovenous malformation (AVM) using AVM embolization. The report could benefit from additional clarity and reorganization of the text to improve its flow and readability. While the introduction is relevant to the case report, it could be more specific to the case being presented.

Minor changes

•         The first time the abbreviature PHIL is included in the abstract and text should be explained.

•         The statement that "all steps to reduce the radiation dose to maximum must be followed with care" (line 23) could be clarified. It is unclear what "maximum" means in this context.

•         In the introduction, organize and present the statistics more clearly.

•         include additional information about the patient's symptoms before diagnosis and any additional diagnostic tests performed. '

•         Provide more information about the interdisciplinary consensus reached before embolization.

•         Provide more detail on the specific risks and challenges associated with treating pregnant patients with bAVMs,

•         Consider adding a discussion of the limitations of the study. Discuss the study's limitations, such as the lack of a comparison group and the single-case design critically.

Author Response

Dear reviewer,

Thank you for your comments, please find answers in the text.

Authors.

  • The first time the abbreviature PHIL is included in the abstract and text should be explained. 

Explanation was added to the text: “PHIL (Precipitating Hydrophobic Injectable Liquid)”

  • The statement that "all steps to reduce the radiation dose to maximum must be followed with care" (line 23) could be clarified. It is unclear what "maximum" means in this context.

The sentence was changed: “​The angiography protocol must be optimized to minimize the radiation dose. “

  • In the introduction, organize and present the statistics more clearly.

The table was added to the text

The main characteristics of brain AVMs are listed in Table 1.

Table 1. Characteristics of brain AVMs.

Brain AVM

Value

Incidence

Approximately 1 per 100,000 per year [1–3]

Prevalence Rate

Approximately 0.01–0.5% [5]

Location

85% vs. 15% for supratentorial (two-thirds superficial and one-third deep) vs. infratentorial [3].

The annual risk of bleeding from unruptured bAVM

Approximately 1–4% [6–7]

The Rebleeding Rate

Is higher, especially in the first year after rupture, ranging from 6% to 15.8% [7]

The potential risk of intracranial hemorrhage compared to a nonpregnant period in women with AVM

Increases 3.27 times during pregnancy and puerperium [10]

Mortality Rate

Maternal mortality rate from rebleeding (up to 28%) and the 14% fetal death rate [5]

  • include additional information about the patient's symptoms before diagnosis and any additional diagnostic tests performed.

All symptoms are described in the text

  • Provide more information about the interdisciplinary consensus reached before embolization.
  • Provide more detail on the specific risks and challenges associated with treating pregnant patients with bAVMs,

A new paragraph was added:

Less than 2% of infants <24 weeks’ gestation and approximately half of the infants from 24 to 27 weeks’ gestation survived to discharge home from the neonatal intensive care unit [60]. Compared to a nonpregnant period in women with AVM, the potential risk of intracranial hemorrhage increases 3.27 times during pregnancy and puerperium [10]. Due to the high rate of rebleeding and the alarming maternal mortality rate from rebleeding (up to 28%) and the 14% fetal death rate, the treatment of ruptured bAVM should not be delayed and must be actively managed [5]. Surgical treatment of posterior fossa AVM is associated with a higher rate of complications. Most posterior fossa AVMs are associated with an increased hemorrhage rate. Thus, they are a predictor of a poor outcome and should be treated even if unruptured to maintain good neurological function [61]. The potential risk of mother blood pressure instability during surgery may negatively affect the fetus. Cerebellar location and deep venous drainage are predictors of hemorrhage. Endovascular embolization is useful and safe for infratentorial AVMs [62].

All the mentioned arguments were discussed in detail during the interdisciplinary meeting.

  • Consider adding a discussion of the limitations of the study. Discuss the study's limitations, such as the lack of a comparison group and the single-case design critically.

 A new paragraph was added:

There are no definitive guidelines on how to properly treat ruptured bAVM during pregnancy. Only a few authors report the radiation dose to the fetus, which is a limit to creating a comparison group. Our center reports only one case. 

Reviewer 2 Report

The article entitled “Embolization of ruptured infratentorial pial AVM in pregnancy” focuses on an arterio-venous neurological malformation characterized by intracranial hemorrhage. This clinical condition is very rare and therefore there are few works in literature. The treatment poses problems  during the pregnancy in terms of management of radiation dose in order to preserve the life of the mother as well as the viability of the fetus. The authors successfully treated the embolization of pinal areteriovenous malformation (AVM) in a primigravida 22-year-old woman  at a gestation of 23 weeks using the transvenous approach and a very low radiation dose (less than 1 µSv) .  The authors state that the patient delivered a healthy baby without congenital disorders. I think that this statement needs to be specified about the need for laboratory/clinical follow-up in the baby or not and the literature possibly would be quoted. This information can make work more apprecciable. Therefore, I think that this article is not suitable for publication in its current version.

Author Response

Dear reviewer,

Thank you for your comments; we changed the formulation in the text.

Authors.

She delivered a baby at 37 weeks of gestation by cesarean section without complications. No congenital disorders were diagnosed by standard screening methods until the age of the newborn was two years.  
